# Sources of COVID-19-Related Information in People with Various Levels of Risk Perception and Preventive Behaviors in Taiwan: A Latent Profile Analysis

**DOI:** 10.3390/ijerph18042091

**Published:** 2021-02-21

**Authors:** Peng-Wei Wang, Yi-Lung Chen, Yu-Ping Chang, Chia-Fen Wu, Wei-Hsin Lu, Cheng-Fang Yen

**Affiliations:** 1Department of Psychiatry, School of Medicine, College of Medicine, Kaohsiung Medical University, Kaohsiung 80708, Taiwan; wistar.huang@gmail.com (P.-W.W.); pino3015@hotmail.com (C.-F.W.); 2Department of Psychiatry, Kaohsiung Medical University Hospital, Kaohsiung 80708, Taiwan; 3Department of Healthcare Administration, Asia University, Taichung 41354, Taiwan; elong@asia.edu.tw; 4Department of Psychology, Asia University, Taichung 41354, Taiwan; 5School of Nursing, The State University of New York, University at Buffalo, New York, NY 14214-3079, USA; yc73@buffalo.edu; 6Department of Psychiatry, Ditmanson Medical Foundation Chia-Yi Christian Hospital, Chia-Yi City 60002, Taiwan

**Keywords:** COVID-19, SARS-CoV-2, risk perception, preventive behavior, information, health belief model

## Abstract

The present study aimed to identify the distinct levels of risk perception and preventive behaviors during the coronavirus disease 2019 (COVID-19) outbreak among people in Taiwan and to examine the roles of information sources in various levels of risk perception and preventive behavior. The online survey recruited 1984 participants through a Facebook advertisement. Their self-reported risk perception, adopted preventive behaviors and COVID-19-related information were collected. We analyzed individuals’ risk perception and adopted preventive behaviors by using latent profile analysis and conducted multinomial logistic regression of latent class membership on COVID-19-related information sources. Four latent classes were identified, including the risk neutrals with high preventive behaviors, the risk exaggerators with high preventive behaviors, the risk deniers with moderate preventive behaviors, and the risk deniers with low preventive behaviors. Compared with the risk neutrals, the risk exaggerators with high preventive behaviors were more likely to obtain COVID-19 information from multiple sources, whereas the risk deniers with moderate preventive behaviors and risk deniers with low preventive behaviors were less likely to obtain COVID-19 information compared with the risk neutrals. Governments and health professions should take the variety of risk perception and adopted preventive behaviors into consideration when disseminating information on COVID-19 to the general public.

## 1. Introduction

Coronavirus disease 2019 (COVID-19) is a highly contagious respiratory infectious disease that has spread rapidly worldwide since the end of 2019 [1]. As of 1 February 2021, roughly 102,895,577 confirmed cases and 2,233,490 deaths have been reported [1]. COVID-19 has challenged modern medicine. Overall hospital mortality from COVID-19 is approximately 15% to 20%, but up to 40% among patients requiring admission to the intensive care units [2]. In addition to physical health, the COVID-19 pandemic has also impacted mental health [3,4], the economy [5], education [6], quality of life [7], occupations [8], and interpersonal relationships [9] of humans. Although two vaccines against COVID-19 have been approved and licensed for general use in the world by the end of 2020 [10], COVID-19 vaccines are not available for people in most countries of the world. Practicing the recommended preventive behaviors—including washing hands, maintaining social distancing, and wearing face masks or coverings—remains the basic and effective method to protect against contracting COVID-19 [11].

### 1.1. Risk Perception and Preventive Behaviors in Coronavirus Disease 2019 Pandemic

A study that reviewed 149 studies on respiratory infectious diseases (RID) concluded that risk perception is the most crucial factor in promoting hygiene and social distancing behaviors [12]. According to the health belief model [13,14], risk perception refers to personal beliefs about the likelihood of suffering a disease [13]. Individuals who perceive a high level of susceptibility to a particular disease will adopt necessary measures to reduce the risk of developing it [15]. Individuals with low perceived susceptibility may deny that they are at risk for contracting a particular illness [15]. There are also people who believe that they are unlikely to suffer from a disease, even as people around them are facing the threat of the same disease; it is very unlikely for them to engage in protective behaviors [15]. A study in the United States demonstrated that engaging in protective behaviors increased as awareness grew of the risk to contracting COVID-19 over the first week of the pandemic [16]. However, risk perception does not guarantee adoption of preventive behaviors during a respiratory infectious disease (RID) pandemic or epidemic [17]. Considering that risk perception has only a partial effect on the adoption of preventive behaviors, public health messages aimed at changing people’s risk perceptions may have only a limited effect in changing people’s preventive behavior [17]. Moreover, individuals with similar levels of risk perception may adopt preventive behaviors of varying degrees. Examining the various levels of risk perception and adoption of preventive behaviors among individuals during an RID pandemic may provide insights into developing strategies for enhancing preventive behaviors.

### 1.2. Roles of Information Sources

It is known that individual characteristics (i.e., demographics), psychosocial factors (i.e., personality, social class, and peer pressure), and structural variables (i.e., knowledge about a disease and experience of contacting with the disease) can affect people’s risk perception of diseases [14,15]. For example, a multinational study revealed that personal experience with COVID-19, individualistic and prosocial values, hearing about COVID-19 from friends and family, trust in government, science, and medical professionals, personal knowledge of government strategy, and personal and collective efficacy were significant predictors of risk perception [18]. Accurate knowledge about a source of infection or disease increases an individual’s perception of risk, which motivates them to change their behavior to reduce risk [19]. Therefore, the present study aimed to examine the roles of information sources in risk perception and preventive behaviors adoption during the COVID-19 pandemic. Information about COVID-19 has been proliferating on traditional and social media since the disease’s outbreak [20]. People may rely on multiple information sources, such as social media, traditional media, websites of government and health agencies, peers, family members, neighbors, and health care workers, during an RID pandemic [21,22,23]. The Internet is a major source of COVID-19 information and has a strong ability to influence its users. A study of 21 countries found that the number of Google searches for “wash hands” increased with the lower speed of the COVID-19 spread [24]. However, systematic analyses and quality assessments revealed that considerable amounts of information on COVID-19 on the Internet often lack scientific rigor [25,26,27,28] and the share of videos contributed by government and health agencies was low [29]. Sources of COVID-19 information may have an impact on risk perception and self-protection behaviors. A latent profile analysis of risk perception and economic confidence in the COVID-19 pandemic revealed that individuals who relied on unofficial information sources were more likely to have high risk perception but low protection efficacy [30]. Whether the sources of information related to COVID-19 factors are associated with various levels of risk perception and the adoption of preventive behaviors warrants further study.

### 1.3. COVID-19 Pandemic and Its Impact in Taiwan

The first COVID-19 case in Taiwan was confirmed on 21 January 2020. During the period from 20 January to 24 February, the Taiwan Centers for Disease Control rapidly produced and implemented a list of at least 124 action items including border control, case identification, quarantine of suspicious cases, proactive case finding, resource allocation, reassurance and education of the public while fighting misinformation, negotiation with other countries and regions, formulation of policies toward schools and childcare, and relief to businesses [31]. With proactive containment efforts and comprehensive contact tracing, the number of COVID-19 cases in Taiwan remained low, as compared with other countries that had widespread outbreaks [32]. Therefore, there was no social lockdown in Taiwan. As of 4 February 2021, Taiwan had tested a total of 358,907 persons showing 919 confirmed cases, of which 9 patients died [1]. However, the pandemic has profoundly affected the economy and unemployment rate in Taiwan [33,34].

### 1.4. Study Aims and Hypotheses

The present study aimed to (1) identify the distinct classes of risk perception and preventive behaviors adopted during the COVID-19 outbreak among people in Taiwan and (2) examine the roles of information sources in these unique classes of risk perception and preventive behavior adoption.

We proposed our hypotheses below. First, according to the health belief model [13,14], risk perception is one but not the sole predictor for adoption of protective behaviors. The study of Sadique et al. also revealed that risk perception does not guarantee adoption of preventive behaviors during an RID pandemic [17]. Therefore, we hypothesized that people with the similar level of risk perception may adopt preventive behaviors to varying degrees. Meanwhile, risk perception may be modified by other factors such as demographics, knowledge of measures, and culture [13,14]. Therefore, we hypothesized that people may perceive varying risks of contracting COVID-19. Second, according to the health belief model [13,14], COVID-19-related information may not only shape people’s knowledge and risk perception but also serve as a cue to adopt preventive behaviors. Therefore, we hypothesized that different sources of COVID-19 information are related to varying risk perception and preventive behaviors.

## 2. Methods

We described the method of recruiting participants, measures and statistical analysis below.

### 2.1. Participants

A total of 2007 participants aged 20 years or older were recruited for an online survey through social media platforms, including Facebook (Facebook, Inc., Menlo Park, CA, USA), LINE (LINE Corporation, Seoul, Korea), and the PTT Bulletin Board System (National Taiwan University, Taiwan), from 20 March 2020 to 5 May 2020. The participants were directed to the research website and were required to respond to the questionnaire voluntarily and anonymously. To collect data from health care workers, we also posted the recruitment information of this research in health care worker groups on Facebook and LINE. A de-duplication protocol was applied to identify multiple submissions to preserve data integrity, including cross-validation of the eligibility criteria of key variables and discrepancies in key data as well as checking for unusually fast completion time (<10 min) [35]. This study was approved by the Institutional Review Board of Kaohsiung Medical University Hospital (KMUHIRB-EXEMPT(I) 20200011).

### 2.2. Measures

#### 2.2.1. Sociodemographics

Sociodemographic data including participants’ age, sex, and education were collected. The age of participants was classified into three groups: <35, 35–49, and ≥50. The education level of participants was classified into three groups: high school or below, bachelor’s degree, and master’s degree and above.

#### 2.2.2. Risk Perception about COVID-19

Risk perception about COVID-19 was measured by asking participants the following five items [36]: (1) whether they would worry about COVID-19 if flu-like symptoms occurred: “If you were to develop flu-like symptoms tomorrow, would you be worried about COVID-19?” (score range: 1–5); (2) whether they had worried about COVID-19: “In the past week, have you ever worried about catching COVID-19?” (score range: 1–5); (3) whether they were currently worried about COVID-19: “Please rate the current level of your worry toward COVID-19” (score range: 1–10); (4) what their anticipated level of worry would be if they were to contract COVID-19: “If you were to develop COVID-19-like symptoms tomorrow, how worried would you be?” (score range: 1–7); and (5) what they perceived their risk of contracting COVID-19 to be compared with other people: “What is the possibility that you catch COVID-19 in a month compared with other people?” (score range: 1–7) (Appendix A). A high score indicated a higher level of worry for or chance of contracting COVID-19. The original questionnaire measuring risk perception about influenza A/H1N1 had acceptable validity and reliability [36]. The results of factor analysis also supported that the COVID-19 version of the questionnaire had acceptable structural validity (factor loading: 0.722–0.897) and internal reliability (Cronbach’s α = 0.712) [37].

#### 2.2.3. Adoption of Preventive Behaviors

The extent to which participants adopted preventive behaviors to protect against COVID-19 was measured by asking participants questions using the following Likert-scale items [36]. Behaviors were (1) avoiding crowded places: “In the past week, did you avoid going to crowded places?”; (2) washing hands: “In the past week, did you wash your hands more often than usual?”; and (3) wearing a mask: “In the past week, did you wear a mask more often than usual?” (Appendix A). These three preventive behaviors were also the most important ones recommended by the Centers for Disease Control and Prevention of the United States to protect people from contracting COVID-19 [11]. The response for each item was a “no” (score 0), “yes, but not due to COVID-19” (score 1), or “yes, due to COVID-19” (score 2).

#### 2.2.4. COVID-19-Related Information Sources

The frequency with which participants obtained COVID-19-related information from the following sources was investigated: Internet media (e.g., Facebook, Twitter, blogs, and Internet news), friends, traditional media (e.g., newspapers, television, and radio broadcasting), academic courses (e.g., online or in-person formal courses lectured by experts), medical staff in health care institutions, coworkers, and family members (Appendix A). The participants were asked how often they obtained information from each information source and were required to respond with never, sometimes, or always. The information sources were classified into low frequency (never and sometimes) and high frequency (always) groups because the sample sizes of the never group were not sufficiently large for most information sources to be categorized as a separate group.

### 2.3. Statistical Analysis

Among the 2007 participants, 6 participants who had missing data on gender, education level, and COVID-19-related information sources were excluded from the analysis. We further excluded 17 participants who reported to be transgender because risk factor analysis is not reliable and robust in this small subgroup. Finally, 1984 participants were included in the analysis.

First, we used latent profile analysis (LPA) to analyze individuals’ risk perception and adoption of preventive behaviors. LPA is a model-based approach that assumes that categorical latent variables account for the covariation between continuous observed variables (indicators). It uses individuals’ responses to indicators to estimate their probability of belonging to a given latent class to identify the latent class to which people most likely belong. We analyzed individuals’ risk perception and adopted preventive behaviors by using the R package *tidyLPA* [37] with the standardization of risk perception and adoption of preventive behaviors. The standard score obtained from the LPA was used to determine the levels of risk perception and adoption of preventive behaviors for latent class membership. The number of classes was selected based on the basic model according to four model fit indices: the Akaike information criterion (AIC), Bayesian information criterion (BIC), entropy, and the bootstrapped likelihood ratio test (BLRT). The model with a lower AIC and BIC had a better fit than that with higher AIC and BIC values. A value of entropy approaching 1 indicates the clear separation of classes [38], and entropy values >0.80 indicate that the latent classes are highly discriminating [39]. For BLRT, *p* < 0.05 indicates that the *k* class model is superior to the *k*-1 class model (*k* represents the number of classes).

Second, to determine the risk factors of latent class determined from the LPA, we conducted multinomial logistic regression of latent class membership on COVID-19-related information sources after adjusting for demographics (age, sex, and education level).

## 3. Results

We listed the results of LPA examining the classes of risk perception and adoption of preventive behaviors and the results of multiple multinomial logistic regression examining the information sources predicting the latent classes below.

### 3.1. Results of LPA

Table 1 presents the correlation matrix with means and standard deviations for the perception of risk of COVID-19 and the adoption of preventive behaviors. Model fit indices for the LPA analysis are shown in Table 2. The four-latent-class model (AIC = 41,009.1, BIC = 41,250.1, and entropy = 0.83) was selected based on its minimal AIC and BIC values and entropy >0.80.

Figure 1 presents the standard score of the perceived risk of COVID-19 and the adoption of preventive behaviors per class. The first—and largest (49.2%, 976/1984, of the sample)—latent class named “risk neutrals with high preventive behaviors” comprised participants with average scores in perceived risk of COVID-19 and the adoption of preventive behaviors. The second (10.3%, 205/1984, of the sample) latent class named “risk exaggerators with high preventive behaviors” consisted of those who had high scores in perceived risk of COVID-19 but average scores for adopting preventive behaviors. The third and fourth classes had low scores in perceived risk of COVID-19 but exhibited various levels of adopting preventive behaviors. The third class named “risk deniers with moderate preventive behaviors” comprised 19.2% (380/1984) of the sample and had average scores for the adoption of preventive behaviors, whereas the fourth class named “risk deniers with low preventive behaviors” consisted of 21.3% (423/1984) of the sample and had low scores for the adoption of preventive behaviors.

### 3.2. Information Sources Predicting the Latent Classes

Figure 2 presents the sociodemographics and COVID-19 information sources among four classes of participants. To determine whether sociodemographics and information sources differed between classes, we conducted a multinomial logistic regression of latent classes on sociodemographics and information sources, with the class of risk neutrals serving as the reference group (Table 3). For sociodemographics with unadjusted multinomial logistic regression, compared with risk neutrals, participants who were early middle-aged (age: 35–49 years) and male were more likely to belong to the risk deniers with low preventive behaviors and risk deniers with moderate preventive behaviors, respectively.

For multiple multinomial logistic regression with adjustment for sociodemographics, compared with the risk neutrals, the risk exaggerators with high preventive behaviors were more likely to obtain COVID-19 information from all types of sources, but only the differences in the information sources of family members, coworkers, and medical staff reached statistical significance. By contrast, the members in the other two classes—the risk deniers with moderate preventive behaviors and risk deniers with low preventive behaviors—were less likely to obtain COVID-19 information from all types of sources compared with risk neutrals. The risk deniers with moderate preventive behaviors were significantly less likely to obtain COVID-19 information from coworkers and friends. Similarly, in addition to coworkers and friends, risk deniers with low preventive behaviors were significantly less likely to obtain COVID-19 information from the Internet media, traditional media, or medical staff.

## 4. Discussion

We listed the discussion regard the classes of risk perception and adoption of preventive behaviors and the roles of information sources predicting the latent classes below.

### 4.1. Classes of Risk Perception and Adoption of Preventive Behaviors

The present study categorized people during the COVID-19 pandemic into four classes according to their levels of risk perception and adoption of preventive behaviors. Compared with the two risk denier groups, both the risk neutrals and risk exaggerators adopted more preventive behaviors, indicating that COVID-19 risk perception may contribute to the adoption of recommended preventive behaviors. However, high perceived risk significantly affects the mental health of people during public health crises [40]. Governments and health professionals should actively promote awareness among the public regarding the threat of COVID-19 without evoking excessive worry. The present study found that 40.5% of participants were classified as risk deniers who adopted fewer preventive behaviors compared with the risk neutrals and risk exaggerators. More than half of them had low scores for adopting preventive behaviors, especially mask wearing.

Although face mask use is beneficial against respiratory infections [41], attitudes toward mask use in the general public and community settings vary across countries [42]. For example, the Surgeon General of the United States advised against healthy people buying masks to preserve the limited supply for professional use in health care settings [42]. The government of Taiwan implemented a mask rationing plan in late January 2020, and people could buy nine masks in 14 days after undergoing verification of their national health insurance cards; thus, the anxiety regarding a mask shortage during the pandemic was reduced [43]. Risk deniers are unlikely to resist wearing masks due to unavailability of masks. Research has revealed that belief in COVID-19-related conspiracy theories is a predictor of resistance to adopting preventive behaviors in the United States [44]. One study suggested that resistance to government advice to wear a mask is also based on the concepts of individualism and distrust of authorities [45]. Further research is warranted to examine the reasons why risk deniers with low protective behavior adoption resist wearing masks.

### 4.2. Information Sources

The present study found that compared with the risk neutrals, the risk exaggerators were more likely to obtain COVID-19 information from all types of sources, whereas the two groups of risk deniers were less likely to obtain COVID-19 information from all types of sources. The cross-sectional design of the study limited the possibility of determining the temporal relationship between information sources and the level of risk perception. A bidirectional relationship may exist between information sources and the level of risk perception. First, people obtaining COVID-19-related information from multiple sources may be more likely to experience infodemic overload and obtain incorrect information, which may exaggerate an unreal sense of the COVID-19 crisis. Moreover, people who refrain from availing themselves of various information sources may underestimate the risk of COVID-19. Second, the risk exaggerators may have a high level of worry related to the COVID-19 pandemic and therefore invest considerable effort and time in urgently gathering information from all information sources. A recent study also found that risk perception may increase people’s risk response and further increase their information-seeking intention and behaviors [46]. By contrast, risk deniers may ignore COVID-19-related information delivered by governments and people or even actively refrain from seeking information from all sources. Access to COVID-19-related information and perceived risk of COVID-19 may reciprocally interact with each other.

The results further indicate that governments and health professionals should consider all population subgroups when disseminating information on COVID-19 to the general public. The information regarding the risk of contracting COVID-19 and preventive strategies should be based on scientific evidence and should be delivered without evoking fear among the public. Furthermore, governments and health professionals should consider people’s interest and motivation and deliver information in a manner acceptable to most people.

### 4.3. Further Studies

There are several issues regarding information, risk perception and protective behaviors during the COVID-19 pandemic that warrant further study. First, according to the health belief model, there might be individual, psychosocial, and structural variables affecting people’s risk perceptions of COVID-19 [13,14]. Knowledge, education level, and socioeconomic status can influence perception of COVID-19 risk in a digital world [47]. Research found that older adults with a lower educational attainment and lower income level were less likely to perceive high risk and stay at home [48]. The present study confirmed that information sources are a structural variable that may influence people’s risk perception and preventive behaviors. Further study is needed to examine the roles of individual, psychosocial, and other structural variables in various classes of risk perception and preventive behaviors.

Second, this study focused on the roles of information sources but did not examine the roles of risk communication for exchanging information. Risk communication refers to the exchange of real-time information, advice, and opinions between experts and people facing threats to their health, economic, or social well-being. The ultimate purpose of risk communication is to enable people at risk to make informed decisions to protect themselves and their loved ones [49]. Research revealed that risk communication has direct and indirect positive effects on preventive behaviors; furthermore, risk perception mediates the relationship between risk communication and preventive behaviors [50]. However, little is known about how people’s health-related behaviors coevolve with social connections for sharing information and discussing urgent pandemic issues [51]. The role of risk communication in various classes of risk perception and protective behaviors during the COVID-19 pandemic warrants further study.

Given that COVID-19 is still not well controlled worldwide [52], healthcare professionals and governments worldwide aspire to control the COVID-19 pandemic through vaccination. Around 20 candidate vaccines are under clinical evaluation [53]. However, the effectiveness of vaccination is limited if people refuse to receive it [54]. Vaccination hesitancy and its relationship with information sources and risk perception should be examined.

### 4.4. Limitations

The present study has several limitations. First, because of the cross-sectional design of this study, no causal inference between information sources, risk perception, and adoption of preventive behaviors can be affirmed; thus, more studies on this topic would be worthwhile. Second, the data were collected starting on 20 March 2020; and, thus, the data did not cover the entire pandemic, which was first reported on 21 January 2020. Third, the participants were recruited from Facebook, LINE, and the PTT Bulletin Board System, which may not completely represent the general population in Taiwan. For example, access to Facebook is not yet universal, and people are not all equally motivated to engage with Facebook [55]. A systematic review on the study recruiting participants via Facebook reported that there appeared to be a bias towards people with a higher educational level [56]. Another review on the study recruiting participants via Facebook reported a bias towards women, young adults, and people with higher education and incomes [57]. Whether our findings could be generalized to the people in the real-world community still requires further investigation. Fourth, although we controlled for the effects of age, gender and educational level, we did not control other possible confounding factors such as sociodemographic status.

## 5. Conclusions

The present study demonstrated that compared with risk deniers, people with a neutral or exaggerated level of risk perception tended to adopt more protective behaviors against COVID-19; meanwhile, over one-fifth of participants who perceived low risk of COVUD-19 adopted fewer preventive behaviors, especially mask wearing. Based on the results, we suggest that governments and health professionals should actively promote awareness among the public regarding the threat of COVID-19. Nevertheless, the programs to promote awareness of COVID-19 risk should avoid evoking excessive worry and damaging mental health. The present study also demonstrated that compared with the risk neutrals, the risk exaggerators tended to obtain COVID-19 information from all types of sources, whereas the risk deniers tended to less obtain COVID-19 information from any source. The results indicated that governments and health professions should take the variety of risk perception and preventive behaviors into consideration when developing prevention programs for COVID-19.

## Figures and Tables

**Figure 1 ijerph-18-02091-f001:**
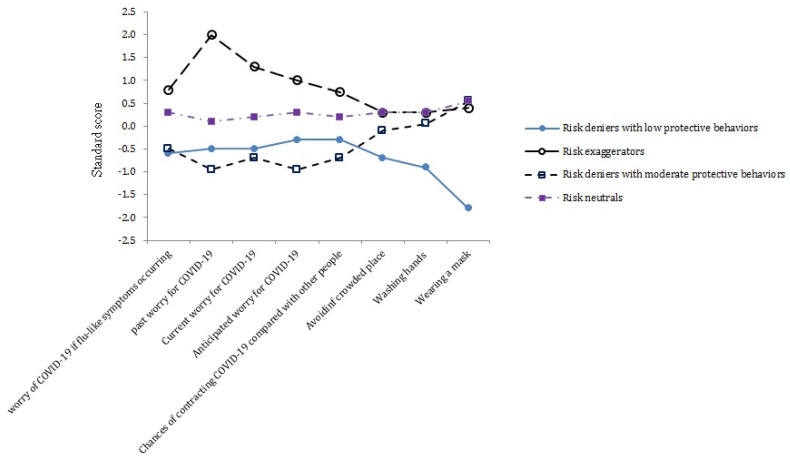
Four classes of participants with various levels of perceived risk of COVID-19 and adoption of preventive behaviors.

**Figure 2 ijerph-18-02091-f002:**
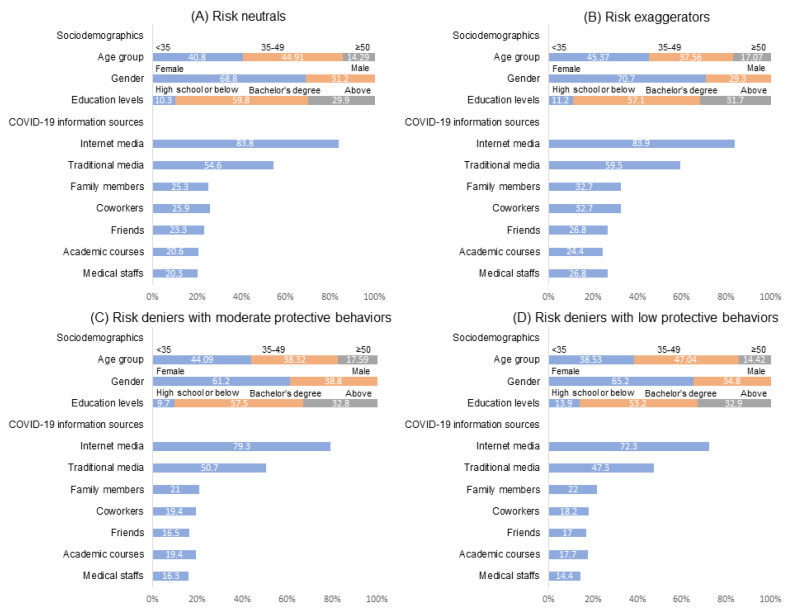
Sociodemographics and COVID-19 information sources between four classes of participants.

**Table 1 ijerph-18-02091-t001:** Descriptive statistics and correlations between study variables (N = 1982).

Variable	1	2	3	4	5	6	7	8
1. Worry of COVID-19 if flu-like symptoms occurring	1							
2. Past worry for COVID-19	0.45	1						
3. Current worry for COVID-19	0.48	0.55	1					
4. Anticipated worry for COVID-19	0.26	0.47	0.37	1				
5. Chances of contracting COVID-19 compared with other people	0.18	0.33	0.23	0.57	1			
6. Avoiding crowded places	0.30	0.21	0.27	0.14	0.08	1		
7. Washing hands	0.25	0.21	0.21	0.10	0.10	0.32	1	
8. Wearing a mask	0.27	0.22	0.25	0.14	0.13	0.33	0.47	1
Mean	2.93	1.59	6.13	2.48	2.53	1.75	1.68	1.66
SD	0.92	1.00	2.25	1.14	1.28	0.55	0.62	0.67

**Table 2 ijerph-18-02091-t002:** Summary of information for selecting the number of latent classes for the latent profile analysis.

No. of Classes	AIC	BIC	Entropy	BLRT (*p*-Value)
1	45,569.5	45,659.2	1.00	<0.01
2	42,451.8	42,591.9	0.98	<0.01
3	41,457.2	41,647.7	0.80	<0.01
4	41,009.1	41,250.1	0.83	<0.01
5	41,127.0	41,418.4	0.78	<0.01
6	40,238.0	40,579.9	0.85	<0.01

Note. AIC = Akaike information criterion, BIC = Bayesian information criterion, BLRT = bootstrapped likelihood ratio test.

**Table 3 ijerph-18-02091-t003:** Multiple multinomial logistic regression of risk perception and adoption of preventive behavior type on demographics and information sources.

Variable	Risk Neutrals with High PB (*n* = 976) *n* (%)	Risk Exaggerators with High PB (*n* = 205) *n* (%)	OR 1	Risk Deniers with Moderate PB (*n* = 380) *n* (%)	OR 2	Risk Deniers with Low PB (*n* = 423) *n* (%)	OR 3
Age ^a^							
<35	397 (40.8)	93 (45.37)	1.00	168 (44.09)	1.00	163 (38.53)	1.00
35–49	437 (44.91)	77 (37.56)	1.33 (0.95–1.85)	146 (38.32)	1.05 (0.72–1.53)	199 (47.04)	1.47 (1.02–2.13)
≥50	139 (14.29)	35 (17.07)	0.93 (0.60–1.44)	67 (17.59)	0.99 (0.61–1.62)	61 (14.42)	1.06 (0.66–1.71)
Gender ^a^							
Female	669 (68.8)	145 (70.7)	1.00	233 (61.2)	1.00	276 (65.2)	1.00
Male	304 (31.2)	60 (29.3)	0.92 (0.66–1.27)	148 (38.8)	1.41 (1.10–1.82)	147 (34.8)	1.18 (0.93–1.49)
Education levels ^a^							
High school or below	100 (10.3)	23 (11.2)	1.00	37 (9.7)	1.00	59 (13.9)	1.00
Bachelor’s degree	582 (59.8)	117 (57.1)	1.03 (0.61–1.75)	219 (57.5)	0.87 (0.57–1.34)	225 (53.2)	1.24 (0.85–1.81)
Master’s degree and above	291 (29.9)	65 (31.7)	0.90 (0.65–1.26)	125 (32.8)	0.88 0.68–1.14)	139 (32.9)	0.81 (0.63–1.05)
COVID-19 information sources (high-frequency) ^b^							
Internet media	815 (83.8)	172 (83.9)	1.02 (0.67–1.54)	302 (79.3)	0.76 (0.56–1.03)	306 (72.3)	0.51 (0.39–0.68)
Traditional media	531 (54.6)	122 (59.5)	1.24 (0.91–1.68)	193 (50.7)	0.87 (0.69–1.11)	200 (47.3)	0.74 (0.59–0.93)
Family members	246 (25.3)	67 (32.7)	1.40 (1.01–1.94)	80 (21.0)	0.79 (0.59–1.05)	93 (22.0)	0.83 (0.63–1.09)
Coworkers	252 (25.9)	67 (32.7)	1.41 (1.02–1.96)	74 (19.4)	0.72 (0.54–0.97)	77 (18.2)	0.64 (0.48–0.86)
Friends	227 (23.3)	55 (26.8)	1.21 (0.86–1.71)	63 (16.5)	0.66 (0.48–0.90)	72 (17.0)	0.68 (0.51–0.91)
Academic courses	200 (20.6)	50 (24.4)	1.27 (0.89–1.82)	74 (19.4)	0.95 (0.70–1.28)	75 (17.7)	0.84 (0.62–1.13)
Medical staff	198 (20.3)	55 (26.8)	1.48 (1.04–2.10)	62 (16.3)	0.79 (0.58–1.09)	61 (14.4)	0.67 (0.49–0.91)

^a^ Crude multinomial logistic regression was conducted. ^b^ Adjusted multiple multinomial logistic regression was conducted after adjustment for sociodemographics. The group of risk neutrals serves as reference in the multinomial logistic regression model. PB: preventive behaviors.

## Data Availability

Restrictions apply to the availability of these data. Only researchers of this study can approach the data.

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
