# Peer review of "Sources of COVID-19-Related Information in People with Various Levels of Risk Perception and Preventive Behaviors in Taiwan: A Latent Profile Analysis"

_ijerph, 2021, doi:10.3390/ijerph18042091_

Round 1

Reviewer 1 Report

Dear Authors,

the impact of the current pandemic is transversal to human nature itself, affecting health and well-being at all levels.

Your contribution proposal needs a finishing touch and some insights in order to deserve publication, since it does not reach sufficient level.

  1. please contextualize better the pandemic scenario;
  2. please better explain the concept of risk perception and which factors could influence it;
  3. why not SARS-CoV-2 among keywords?
  4. the questionnaire administered is tailor-made, right? If so, does it contain parts of validated questionnaires by far? Please provide detailed informations;
  5. in such a peculiar scenario you should consider other factors: what about risk stress, PTSD, burnout, stigma and discrimination, FOMO?! You have to deal with other psi-factors;
  6. figure 1 is too small and too grey;
  7. what about SES? deal with it;
  8. conclusion must be improved. What are the possible repercussions? What suggestions to give to the health policy maker? Define a clear "take home message" from your perspective and address a conclusion section. You need conclusions;
  9. Please state in the conclusion if you will re-contact participants to retake the questionnaire after the pandemic or after they get vaccine;
  10. what about vaccine hesitancy in this scenario? deal with it, even because CoViD-19 vaccines are now available all over the globe (please also refer to https://www.who.int/news-room/spotlight/ten-threats-to-global-health-in-2019 ) and it's crucial in risk perception

You need to significantly improve the manuscript in order to deserve publication.

Please update these gaps referring to the following references:

  • Irigoyen-Camacho, M.E.; Velazquez-Alva, M.C.; Zepeda-Zepeda, M.A.; Cabrer-Rosales, M.F.; Lazarevich, I.; Castaño-Seiquer, A. Effect of Income Level and Perception of Susceptibility and Severity of COVID-19 on Stay-at-Home Preventive Behavior in a Group of Older Adults in Mexico City. Int. J. Environ. Res. Public Health 2020, 17, 7418
  • Baldassarre, A.; Giorgi, G.; Alessio, F.; Lulli, L.G.; Arcangeli, G.; Mucci, N. Stigma and Discrimination (SAD) at the Time of the SARS-CoV-2 Pandemic. Int. J. Environ. Res. Public Health 2020, 17, 6341
  • Sarah Dryhurst, Claudia R. Schneider, John Kerr, Alexandra L. J. Freeman, Gabriel Recchia, Anne Marthe van der Bles, David Spiegelhalter & Sander van der Linden (2020) Risk perceptions of COVID-19 around the world, Journal of Risk Research, DOI: 10.1080/13669877.2020.1758193
  • Wise, T., et al. (2020) Changes in risk perception and self-reported protective behaviour during the first week of the COVID-19 pandemic in the United States. Royal Society Open Science. doi.org/10.1098/rsos.200742

Dear Authors,

your contribution results interesting but still poor. Deal with hints given and try to deserve publication.

Author Response

Reviewer 1

Thank you so much for your valuable comments on out manuscript. We have revised our manuscript based on reviewers’ suggestions. Please let me know if we need to provide anything else regarding this revision.

Comment 1

Please contextualize better the pandemic scenario.

Response

Thank you for your suggestion. We revised the pandemic scenario in the beginning of Introduction as below.

“Coronavirus disease 2019 (COVID-19) is a highly contagious respiratory infectious disease that has spread rapidly worldwide since the end of 2019 [1]. As of February 1, 2021, roughly 102,895,577 confirmed cases and 2,233,490 deaths have been reported [1]. COVID-19 has challenged modern medicine. Overall hospital mortality from COVID-19 is approximately 15% to 20%, but up to 40% among patients requiring admission to the intensive care units [2]. In addition to physical health, the COVID-19 pandemic has also impacted mental health [3, 4], the economy [5], education [6], quality of life [7], occupations [8], and interpersonal relationships [9] of humans.”

Comment 2

please better explain the concept of risk perception and which factors could influence it.

Response

“According to the health belief model [13, 14], risk perception refers to personal beliefs about the likelihood of suffering a disease [13]. Individuals who perceive a high level of susceptibility to a particular disease will adopt necessary measures to reduce the risk of developing this disease [15]. Individuals with low perceived susceptibility may deny that they are at risk for contracting a particular illness [15]There are also people who believe that they are unlikely to suffer from a disease even people around are facing the threat of this disease; it is very unlikely for them to engage in preventive behaviors [15]. A study in the United States demonstrated that engaging in preventive behaviors increased as growing awareness of risk to contracting COVID-19 over the first week of the pandemic [16].

“It is known that individual characteristics (i.e., demographics), psychosocial factors (i.e., personality, social class, and peer pressure), and structural variables (i.e., knowledge about a disease and experience of contacting with the disease) can affect people’s risk perception of diseases [14, 15]. For example, a multinational study revealed that personal experience with COVID-19, individualistic and prosocial values, hearing about COVID-19 from friends and family, trust in government, science, and medical professionals, personal knowledge of government strategy, and personal and collective efficacy were significant predictors of risk perception [18]. Accurate knowledge about a source of infection or disease increases an individual’s perception of risk, which motivates them to change their behavior to reduce risk [19]. Therefore, the present study aimed to examine the roles of information sources in risk perception and preventive behaviors adoption during the COVID-19 pandemic.”

Comment 3

why not SARS-CoV-2 among keywords?

Response

Thank you for your reminding. We added it into Keywords.

Comment 4

the questionnaire administered is tailor-made, right? If so, does it contain parts of validated questionnaires by far? Please provide detailed informations.

Response

We added the results of examining psychometric of the questionnaire for risk perception and preventive behaviors into the revised manuscript as below.

“The results of factor analysis also supported that the COVID-19 version of the questionnaire had acceptable structural validity (factor loading: 0.722-0.897) and internal reliability (Cronbach's α = 0.712) [37].

“…These three preventive behaviors were also the most important ones recommended by the Centers for Disease Control and Prevention of the United States to protect people from contracting COVID-19 [11].

Comment 5

in such a peculiar scenario you should consider other factors: what about risk stress, PTSD, burnout, stigma and discrimination, FOMO?! You have to deal with other psi-factors.

Response

Thank you for your comment. We agreed that there might be individual, psychosocial, and structural variables affect people’s risk perceptions of COVID-19. Information sources are a structural variable examined in this study. We supplemented the contents below to emphasize the importance of further study to examined the roles of individual, psychosocial, and other structural variables in various classes of risk perception and preventive behaviors.

Introduction

“The present study aimed to examine the roles of information sources in risk perception and preventive behaviors adoption during the COVID-19 pandemic.”

Discussion

“4.3. Further studies

There are several issues regarding information, risk perception and preventive behaviors during the COVID-19 pandemic warrants further study. First, according to the health belief model, there might be individual, psychosocial, and structural variables affect people’s risk perceptions of COVID-19 [13, 14]. Knowledge, education level, and socioeconomic status can influence perception of COVID-19 risk in a digital world [47]. Research found that older adults with a lower educational attainment and lower income level were less likely to perceive high risk and stay at home [48]. The present study confirmed that information sources are a structural variable that may influence people’s risk perception and preventive behaviors. Further study is needed to examined the roles of individual, psychosocial, and other structural variables in various classes of risk perception and preventive behaviors.”

Comment 6

figure 1 is too small and too grey.

Response

Thank you for your suggestion. We revised Figure 1 to make it bigger and clearer.

Comment 7

what about SES? deal with it.

Response

Thank you for your comment. Although education level might partially represent participants’ sociodemographic status (SES), we did not use a formal indicator to survey participants’ SES. In addition to adding it as one of issues warranted further study (described above), we listed it as one of limitations of this study in the revised manuscript.

“Fourth, although we controlled for the effects of age, gender and educational level, we did not control other possible confounding factors such as sociodemographic status.”

Comment 8

conclusion must be improved. What are the possible repercussions? What suggestions to give to the health policy maker? Define a clear "take home message" from your perspective and address a conclusion section. You need conclusions.

Response

Thank you for your comment. We revised Conclusion section to make it clear as below.

“The present study demonstrated that compared with risk deniers, people with a neutral or exaggerated level of risk perception tended to adopt more preventive behaviors against COVID-19; meanwhile, over one-fifth of participants who perceived low risk of COVUD-19 adopted fewer preventive behaviors, especially mask wearing. Based on the results, we suggest that governments and health professionals should actively promote awareness among the public regarding the threat of COVID-19. Nevertheless, the programs to promote awareness of COVID-19 risk should avoid evoking excessive worry and damaging mental health. The present study also demonstrated that compared with the risk neutrals, the risk exaggerators tended to obtain COVID-19 information from all types of sources, whereas the risk deniers tended to obtain COVID-19 information from all types of source. The results indicated that governments and health professions should take the variety of risk perception and adoption of preventive behaviors into consideration when developing prevention programs for COVID-19.

Comment 9

Please state in the conclusion if you will re-contact participants to retake the questionnaire after the pandemic or after they get vaccine.

Response

As we have described in “2.1. Participants”, participants respond to the questionnaire anonymously. It is impossible to re-contact the same participants to retake the questionnaire after the pandemic or after they get vaccine. Nevertheless, we agreed that it is important to evaluate people’s attitude toward vaccination for COVID-19, information sources of vaccination, and the changes of risk perception after getting vaccine. We listed it as one of issues warranted further study as described in the response to Comment 5.

Comment 10

what about vaccine hesitancy in this scenario? deal with it, even because CoViD-19 vaccines are now available all over the globe (please also refer to https://www.who.int/news-room/spotlight/ten-threats-to-global-health-in-2019 ) and it's crucial in risk perception.

Response

We agree that vaccination hesitancy is a very important issue in COVID-19 pandemic. We added it as one of issues warranted further study into the revised manuscript as below.

“Given that COVID-19 are still not well controlled worldwide in the past one more year [52], healthcare professionals and governments worldwide aspire after vaccinations to control the COVID-19 pandemic. Around 20 candidate vaccines are under clinical evaluation [53]. However, the effectiveness of vaccination is limited if people refuse uptake [54]. Vaccination hesitancy and its relationship with information sources and risk perception should be examined.”

Comment 11

Please update these gaps referring to the following references:

  • Irigoyen-Camacho, M.E.; Velazquez-Alva, M.C.; Zepeda-Zepeda, M.A.; Cabrer-Rosales, M.F.; Lazarevich, I.; Castaño-Seiquer, A. Effect of Income Level and Perception of Susceptibility and Severity of COVID-19 on Stay-at-Home Preventive Behavior in a Group of Older Adults in Mexico City. Int. J. Environ. Res. Public Health 2020, 17, 7418
  • Baldassarre, A.; Giorgi, G.; Alessio, F.; Lulli, L.G.; Arcangeli, G.; Mucci, N. Stigma and Discrimination (SAD) at the Time of the SARS-CoV-2 Pandemic. Int. J. Environ. Res. Public Health 2020, 17, 6341
  • Sarah Dryhurst, Claudia R. Schneider, John Kerr, Alexandra L. J. Freeman, Gabriel Recchia, Anne Marthe van der Bles, David Spiegelhalter & Sander van der Linden (2020) Risk perceptions of COVID-19 around the world, Journal of Risk Research, DOI: 10.1080/13669877.2020.1758193
  • Wise, T., et al. (2020) Changes in risk perception and self-reported protective behaviour during the first week of the COVID-19 pandemic in the United States. Royal Society Open Science. doi.org/10.1098/rsos.200742

Response

Thank you for your suggestion. We added these references into the revised manuscript as below.

  1. “Research found that older adults with a lower educational attainment and lower income level were less likely to perceive high risk and stay at home [48].”
  2. “Knowledge, education level, and socioeconomic status can influence perception of COVID-19 risk in a digital world [47].”
  3. “For example, amultinational study revealed that personal experience with COVID-19, individualistic and prosocial values, hearing about COVID-19 from friends and family, trust in government, science, and medical professionals, personal knowledge of government strategy, and personal and collective efficacy were significant predictors of risk perception [18].”
  4. “ A study in the United States demonstrated that engaging in preventive behaviors increased as growing awareness of risk to contracting COVID-19 over the first week of the pandemic [16].”

Reviewer 2 Report

Dear Authors,
This is an interesting work.
I recommend you to include as many keywords as possible to position the paper (Sars-cov2), coronavirus, .....

You should include a section with the research questions or hypotheses you hope to refute.

I would recommend you to add a data visualization in the introduction in order to show the covid-19 pandemic and its impact in Taiwan.

I would also recommend that you include a data visualization or infographic related to all the proposed data collection at a schematic level and also illustrate the sample visually in the results.

You should include a section on future lines of research.

You should include in supplementary information /appendix an example of the survey you sent to the participants.

Best memories.

Author Response

Reviewer 2

Thank you so much for your valuable comments on out manuscript. We have revised our manuscript based on reviewers’ suggestions. Please let me know if we need to provide anything else regarding this revision.

Comment 1

I recommend you to include as many keywords as possible to position the paper (Sars-cov2), coronavirus.

Response

Thank you for your reminding. We added “SARS-CoV-2” and “health belief model” into Keywords.

Comment 2

You should include a section with the research questions or hypotheses you hope to refute.

Response

Thank you for your suggestion. We described our study aims and hypotheses in section 1.4. as below.

“1.4. Study Aims and Hypotheses

The present study aimed to (1) identify the distinct classes of risk perception and preventive behaviors adopted during the COVID-19 outbreak among people in Taiwan and (2) examine the roles of information sources in these unique classes of risk perception and preventive behavior adoption.

We proposed our hypotheses below. First, according to the health belief model [13, 14], risk perception is one but not the sole predictor for adoption of preventive behaviors. The study of Sadique et al. also revealed that risk perception does not guarantee adoption of preventive behaviors during an RID pandemic [17]. Therefore, we hypothesized that people with the similar level of risk perception may adopt preventive behaviors to varying degrees. Meanwhile, risk perception may be modified by other factors such as demographics, knowledge of measures, and culture [13, 14]. Therefore, we hypothesized that people may perceive varying risks of contracting COVID-19. Second, according to the health belief model [13, 14], COVID-19-related information may not only shape people’s knowledge and then risk perception but also serve as a cue to adopt preventive behaviors. Therefore, we hypothesized that different sources of COVID-19 information are related to varying risk perception and preventive behaviors.

Comment 3

I would recommend you to add a data visualization in the introduction in order to show the covid-19 pandemic and its impact in Taiwan.

Response

Thank you for your suggestion. We added a paragraph to describe the COVID-19 pandemic and its impact in Taiwan as below.

“1.3. Covid-19 pandemic and its impact in Taiwan

The first COVID-19 case in Taiwan was confirmed on 21 January 2020. During the period from January 20 to February 24, the Taiwan Centers for Disease Control rapidly produced and implemented a list of at least 124 action items including border control, case identification, quarantine of suspicious cases, proactive case finding, resource allocation, reassurance and education of the public while fighting misinformation, negotiation with other countries and regions, formulation of policies toward schools and childcare, and relief to businesses [31]. With proactive containment efforts and comprehensive contact tracing, the number of COVID-19 cases in Taiwan remained low, as compared with other countries that had widespread outbreaks [32]. Therefore, there was no social lockdown in Taiwan. As of 4 February 2021, Taiwan had tested a total of 358,907 persons showing 919 confirmed cases, of which 9 patients died [1]. However, the pandemic has profoundly affected the economy and unemployment rate in Taiwan [33, 34].”

Comment 4
I would also recommend that you include a data visualization or infographic related to all the proposed data collection at a schematic level and also illustrate the sample visually in the results.

Response

Thank you for your suggestion. We added Figure 2 to illustrate the distributions of demographics and information sources among four classes of participants with various risk perception and adoption of preventive behaviors. Please refer to Figure 2.

“Figure 2 presents the distributions of sociodemographics and COVID-19 information sources among four classes of participants.”

Comment 5
You should include a section on future lines of research.

Response

Thank you for your suggestion. We added a paragraph to describe future studies as below.

“4.3. Further studies

There are several issues regarding information, risk perception and preventive behaviors during the COVID-19 pandemic warrants further study. First, according to the health belief model, there might be individual, psychosocial, and structural variables affect people’s risk perceptions of COVID-19 [13, 14]. Knowledge, education level, and socioeconomic status can influence perception of COVID-19 risk in a digital world [47]. Research found that older adults with a lower educational attainment and lower income level were less likely to perceive high risk and stay at home [48]. The present study confirmed that information sources are a structural variable that may influence people’s risk perception and preventive behaviors. Further study is needed to examined the roles of individual, psychosocial, and other structural variables in various classes of risk perception and preventive behaviors.

Second, this study focused on the roles of information sources but did not examine the roles of risk communication for exchanging information. Risk communication refers to the exchange of real-time information, advice and opinions between experts and people facing threats to their health, economic or social well-being. The ultimate purpose of risk communication is to enable people at risk to take informed decisions to protect themselves and their loved ones [49]. Research revealed that risk communication has direct and indirect positive effects on preventive behaviors; furthermore, risk perception mediates the relationship between risk communication and preventive behaviors [50]. However, little is known about how people’s health-related behaviors coevolve with social connections for sharing information and discussing urgent pandemic issues [51]. The role of risk communication in various classes of risk perception and preventive behaviors during the COVID-19 pandemic warrants further study.

Given that COVID-19 are still not well controlled worldwide in the past one more year [52], healthcare professionals and governments worldwide aspire after vaccinations to control the COVID-19 pandemic. Around 20 candidate vaccines are under clinical evaluation [53]. However, the effectiveness of vaccination is limited if people refuse uptake [54]. Vaccination hesitancy and its relationship with information sources and risk perception should be examined.

Comment 6
You should include in supplementary information /appendix an example of the survey you sent to the participants.

Response

Thank you for your suggestion. We added Supplementary Table S1 to describe the measures used in this study.

Reviewer 3 Report

  1. This study conducted an online survey through a Facebook advertisement and has a very high potential for selection bias. How this kind of selection bias could be controlled in this study should be explained. Furthermore, how the authors detected and checked whether the respondents to this online survey self-reported correct of fake information on themselves (for example, education, age, etc.) should be explained.
  2. This study focuses on the roles of information sources but does not pay attention to roles of risk communication for exchanging information. Literature Review part needs to cover the relationship between risk communication and preventive behaviours even briefly by adding these two literatures as below:

Heydari, S.T., Zarei, L., Sadati, A.K. et al. The effect of risk communication on preventive and protective Behaviours during the COVID-19 outbreak: mediating role of risk perception. BMC Public Health 21, 54 (2021). 

Lim S, Nakazato H. The Emergence of Risk Communication Networks and the Development of Citizen Health-Related Behaviors during the COVID-19 Pandemic: Social Selection and Contagion Processes. International Journal of Environmental Research and Public Health. 2020; 17(11): 4148.

  1. In 1.3. Study Aim, there are two hypotheses developed without any theoretical backgrounds. Some theoretical reasonings for these two hypotheses are required.

Author Response

Reviewer 3

Thank you so much for your valuable comments on out manuscript. We have revised our manuscript based on reviewers’ suggestions. Please let me know if we need to provide anything else regarding this revision.

Comment 1

This study conducted an online survey through a Facebook advertisement and has a very high potential for selection bias. How this kind of selection bias could be controlled in this study should be explained. Furthermore, how the authors detected and checked whether the respondents to this online survey self-reported correct of fake information on themselves (for example, education, age, etc.) should be explained.

Response

Thank you for your comment. We added the description for the de-duplication protocol that was applied in the present study to identify multiple submissions to preserve data integrity as below. Meanwhile, we expanded the paragraph describing the limitation resulted from the online survey through a Facebook advertisement as below.

“A de-duplication protocol was applied to identify multiple submissions to preserve data integrity, including cross-validation of the eligibility criteria of key variables and discrepancies in key data and checking for unusually fast completion time (< 10 minutes) [35].”

Third, the participants were recruited from Facebook, LINE, and the PTT Bulletin Board System, which may not completely represent the general population in Taiwan. For example, access to Facebook is not yet universal, and people are not all equally motivated to engage with Facebook [55]. A systematic review on the study recruiting participants via Facebook reported that there appeared to be a bias towards people with higher education [56]. Another review on the study recruiting participants via Facebook reported a bias towards women, young adults, and people with higher education and incomes [57]. Whether our findings could be generalized to the people in the real-world community still requires further investigation.

Comment 2

This study focuses on the roles of information sources but does not pay attention to roles of risk communication for exchanging information. Literature Review part needs to cover the relationship between risk communication and preventive behaviours even briefly by adding these two literatures as below:

Heydari, S.T., Zarei, L., Sadati, A.K. et al. The effect of risk communication on preventive and protective Behaviours during the COVID-19 outbreak: mediating role of risk perception. BMC Public Health 21, 54 (2021). 

Lim S, Nakazato H. The Emergence of Risk Communication Networks and the Development of Citizen Health-Related Behaviors during the COVID-19 Pandemic: Social Selection and Contagion Processes. International Journal of Environmental Research and Public Health. 2020; 17(11): 4148.

Response

Thank you for your suggestion. We added risk communication as one of issues warranted further study in the revised manuscript as below.

“Second, this study focused on the roles of information sources but did not examine the roles of risk communication for exchanging information. Risk communication refers to the exchange of real-time information, advice and opinions between experts and people facing threats to their health, economic or social well-being. The ultimate purpose of risk communication is to enable people at risk to take informed decisions to protect themselves and their loved ones [49]. Research revealed that risk communication has direct and indirect positive effects on preventive behaviors; furthermore, risk perception mediates the relationship between risk communication and preventive behaviors [50]. However, little is known about how people’s health-related behaviors coevolve with social connections for sharing information and discussing urgent pandemic issues [51]. The role of risk communication in various classes of risk perception and preventive behaviors during the COVID-19 pandemic warrants further study.”

Comment 3

In 1.3. Study Aim, there are two hypotheses developed without any theoretical backgrounds. Some theoretical reasonings for these two hypotheses are required.

Response

Thank you for your comment. We revised the hypothesis section according to the health belief model below.

“We proposed our hypotheses below. First, according to the health belief model [13, 14], risk perception is one but not the sole predictor for adoption of preventive behaviors. The study of Sadique et al. also revealed that risk perception does not guarantee adoption of preventive behaviors during an RID pandemic [17]. Therefore, we hypothesized that people with the similar level of risk perception may adopt preventive behaviors to varying degrees. Meanwhile, risk perception may be modified by other factors such as demographics, knowledge of measures, and culture [13, 14]. Therefore, we hypothesized that people may perceive varying risks of contracting COVID-19. Second, according to the health belief model [13, 14], COVID-19-related information may not only shape people’s knowledge and then risk perception but also serve as a cue to adopt preventive behaviors. Therefore, we hypothesized that different sources of COVID-19 information are related to varying risk perception and preventive behaviors.”

Round 2

Reviewer 1 Report

Dear authors,

thanks for addressing reviewers' suggestions.

You managed to improve your manuscript proposal, that deserves publication IMHO.

Best,

Author Response

Thank you for your comment.

Reviewer 2 Report

Dear authors,
You made severe changes in the document and it shows the effort and the well done work. Forward. 

Author Response

Thank you for your comment.

Reviewer 3 Report

The authors have reflected the comments and suggestions from this reviewer as much as possible. This revision is satisfactory. 

Author Response

Thank you for your comment.